# Graph convolutional neural networks with uncertainty modelling applied to edge detection in mammograms

Anonymous Full Paper
Submission 41

## Abstract

This paper addresses the use of a graph convolutional network for delineation of structures and estimation of landmarks in mammograms, a critical step in the evaluation of image quality in breast cancer screening. In this context, the ability to estimate the uncertainty of the predicted positions is crucial. In the current work we focus on the pectoral muscle, where the variability in muscle visibility across images introduces significant uncertainty. Our main contribution is a novel modification of a deep graph convolutional network (GCN) that not only locates key points along the muscle boundary but also provides uncertainty estimates, which are useful for selecting images that must be evaluated by a human. We introduce a novel approach to estimate both aleatoric and epistemic uncertainties using a GCN framework. Aleatoric uncertainty captures variability in ground truth due to annotator differences, while epistemic uncertainty accounts for the model's inherent limitations. Our method was tested on in-house annotated mammograms and the external InBreast dataset, demonstrating comparable accuracy to human annotators and robustness in the presence of domain shifts. The uncertainty estimates were found to be highly accurate, confirming their potential for identifying cases that require human review.

## 1 Introduction

Breast cancer is the most common form of cancer in females, worldwide, and mammographic screening is an effective way of detecting cancers at an early stage.

In mammography screening it is crucial to maintain high image quality to ensure the best possible visualization of breast tissue and the identification of potential breast cancer indicators. This includes technical quality in the form of image sharpness and contrast as well as the proper positioning, which ensures that the relevant parts of the breast are depicted. The quality of the images significantly impacts the rates of patient recall and the detection of cancer through screening, thus affecting the accuracy and reliability of the screening process. Therefore, mammograms routinely undergo quality assessment by radiographers, which would benefit

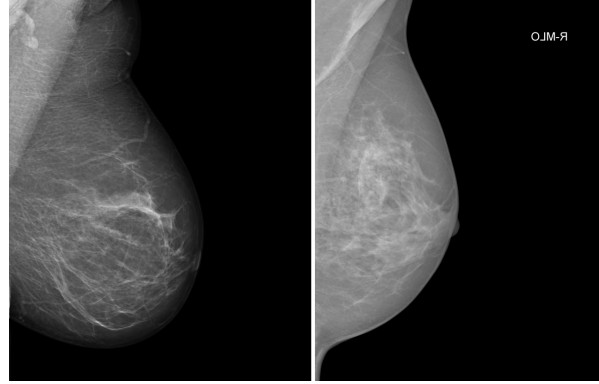

**Figure 1.** Sample mammograms where the pectoral muscle is clearly depicted (left) and blurry (right).

from automatic analysis.

An important part of the quality assessment concerns the depiction of the pectoral muscle in the mediolateral oblique (MLO) view, relating to its size, its shape, and its orientation (Waade et al. [1]). The muscle should be located in the top left (right) corner of a standard X-ray mammogram in MLO-view. The complexity of the task varies considerably, as shown in Figure 1, where the muscle is clearly visible as a light area in the left image, while the right image shows a more difficult case where the lower part of the muscle is blurry.

Previous studies have used different methods for segmenting the muscle, using various proprietary versions of convolutional neural nets (CNNs). Ma et al. [2] apply a method similar to a U-Net and concludes it works better than traditional imaging techniques. Brahim et al. [3] use a CNN model heatmaps generated by a GradCam method. PeM-Net (Yu et al. [4]) uses a InceptionResNetv2 backbone and a complex up-sampling scheme to generate pixel masks. Guo et al. [5] use a U-Net to segment the muscle. Yang et al. [6] uses deep learning in the way of a modified U-Net model to segment pectoral muscle volume from computed tomography images.

While these methods may provide state-of-the-art segmentation masks, their handling of uncertainty leaves something to be desired, as they focus only on average error (over different datasets and different criteria). They do not provide explicit probability distributions or confidence intervals for their location estimates. We argue that the application to image quality control would benefit from such estimates,

so that a human could perform the evaluations in critical cases where the predicted locations are likely to be substantially off target.

Graph convolutional neural nets (GCNs) have been introduced as an alternative to heatmap based methods for landmark detection to better capture relationships between landmarks. The aim is to enable the model to learn relations between point positions and thereby improve on the relative positioning of the landmarks. In the present study, we use the GCN approach to identify points along the border of the muscle. Since the muscle is located in a corner, the task of identifying the border that extends from the top edge to the side edge is equivalent to segmenting it. However, landmark based modelling extends directly to locating of singular key points that are used for quality assessment of mammogram, such as the nipple and the infra-mammary fold. While these are not included in the present study, this is a natural extension of the landmark approach, which can also benefit from learning the geometric relations among the points. While heatmap based methods can also be used for these tasks, the location of singular points is handled more directly in landmark based methods.

We adopt a modification of the "Deep Adaptive Graph" (DAG) framework, which has previously been used for identifying key points in faces (Li et al. [7]). Our contribution is an extension of the DAG framework that provides an explicit probabilistic model for the location of the key points. We train the model to output estimates of the aleatoric (truly random) uncertainties together with the key point coordinates.

To our knowledge, no attempts to explicit uncertainty modelling in GCNs have been published to date, and such modelling has been called for: "To date, the vast majority of existing works do not take into account the uncertainty of a GCN regarding its prediction, which is alarming especially in high-stake scenarios." (Kang et al. [8]). We use the Laplace distribution, which has been used as a prior for Bayesian uncertainty estimation in a general context (Kendall and Gal [9]), but to our knowledge not in a GCN setting.

In addition we estimate the epistemic (model related) uncertainties through the variations within a model ensemble, and combine the aleatoric and epistemic uncertainties to create accurate confidence intervals.

## 2  Method

We include only an informal description of the "Deep Adaptive Graph" (DAG) framework (Li et al. [7]), which we build on, and refer to this source for technical details. The model's basic task is to locate a vector of points $v \in \mathbb{R}^{n \times 2}$ in an image as close

as possible to the ground truth of correct locations $v^*$. It uses a high-resolution convolutional net (HR-Net) to generate a feature map for the given image. These features are fed into a GCN together with the geometry of a current location estimate $v$ and give a vector $\Delta v$ as output. The process starts with an initial point vector $v_0$, which is updated iteratively: $v_{i+1} = v_i + \Delta v_i$, which is intended to move toward $v^*$. The algorithm described by (Li et al. [7]) also included a so-called global step prior to the $\Delta v$ updates, but this was not included in the present application. We let $T$ be the number of local steps and for convenience define $\mu = v_T$.

## 2.1  Aleatoric uncertainty estimation

By aleatoric uncertainty we mean uncertainty about the ground truth that is "truly random" in the sense that it cannot be eliminated by any amount of training data or any kind of model (Hüllermeier and Waegeman [10]). It represents the randomness among different annotators - or even the same annotator at different times - in how they place the markings. In the right image of Figure 1, e.g. the delineation of the lower part of the muscle is likely to vary substantially in this way.

Earlier research on aleatoric uncertainty in GCN models focus on labelling problems, where the task is to assign properties to the nodes in a graph (Vashishth et al. [11]). A recent overview of uncertainty in GCN models is given in (Wang et al. [12]), which gives a taxonomy of types and sources for uncertainties, and ways to estimate them. Our work is different in the way that we develop an architecture that allows the model to estimate 2-dimensional locations in parallel with uncertainty estimates for the same locations. We accomplish this by extending the method of Li et al with the inclusion of a separate GCN module that estimates aleatoric uncertainties through parameterized random distributions. It is structurally equivalent to the GCN module that computes $\Delta v$ and is applied after the last iteration $T$. It takes as input the HR-net features evaluated at the locations $\mu$ together with the geometric features of $\mu$. The output is denoted by $\log(b) \in \mathbb{R}^{n \times 2}$, where $b_x^i$ and $b_y^i$ represent the uncertainties for $\mu_x^i$ $\mu_y^i$.

Rather than the traditional deep learning approach of defining a loss function which measures the distance from the desired outcome, we view the entire model as a parameterized statistical model of the training data. For each point in $v^{*i}$, we assume that $v_x^{*i}$ has the Laplace (double exponential) distribution with median $\mu_x^i$ scale parameter $b_x^i$, which we write $v_x^{*i} \sim \mathcal{L}(\mu_x^i, b_x^i)$ and similarly $v_y^{*i} \sim \mathcal{L}(\mu_y^i, b_y^i)$. We assumed all components to be

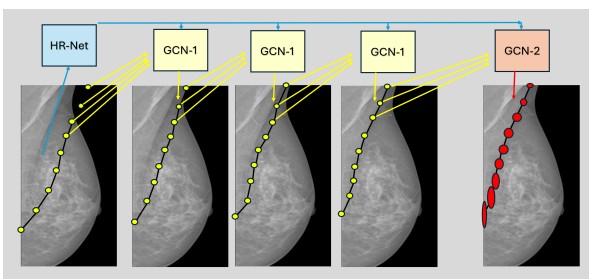

**Figure 2.** Computational pipeline.

independent, which gives the following likelihood:

$$L = \prod_k^M \prod_i^n \prod_{j \in \{x,y\}} \frac{1}{2b_j^{k,i}} \exp\left( -\frac{|v_j^{*k,i} - \mu_j^{k,i}|}{b_j^{k,i}} \right)$$

Here, $k$ runs over the $M$ training images, $i$ runs over the $n$ key points and $j$ runs over the $x$ and $y$ dimension.

Our approach is to optimize the model parameters with respect to this likelihood function, but as usual in statistical modelling, we minimize $-\log(L)$ instead, which has the same optimum and more favorable numerical properties:

$$-\log L = \sum_k^M \sum_i^n \sum_{j \in \{x,y\}} \log(2b_j^{k,i}) + \frac{1}{b_j^{k,i}}|v_j^{*k,i} - \mu_j^{k,i}|$$

Minimizing this expression amounts to training the model end-to-end to estimate locations $\mu_x^i, \mu_y^i$ and uncertainties $b_x^i, b_y^i$ for a given input image. To our knowledge, this approach to estimating aleatoric uncertainties for GCN models is novel.

We also considered using normal distributions instead of Laplace, but chose the latter because it is the natural generalization of the L1-error that was used by (Li et al. [7]). It has the property that it penalizes deviations linearly rather than quadratically, which makes it more robust. Normal distributions might place too much emphasize on the cases where the model has trouble reproducing the ground truth.

### 2.1.1 Model pipeline

The model pipeline is shown in Figure 2. The HR-Net (blue color) computes localized features, which are fed into the GCN models. GCN-1 (yellow) reads the HR-Net features at the initial points, combined with their geometry, and moves the points (hopefully) toward the correct locations. This process is repeated $T$ times. Then GCN-2 (red) takes the HR-Net features of these final locations as input, together with their geometry, and outputs the estimated x- and y- uncertainties, illustrated with red ellipses. The entire model is trained end-to-end to optimize the Laplace log-likelihood of the ground truth data.

## 2.2 Epistemic uncertainty estimation

Our approach for estimating epistemic (model related) uncertainty (Hüllermeier and Waegeman [10]) is more standard and straight forward. We train a set of models with cross-validation, and use them as an ensemble when evaluating the test sets (Dutschmann et al. [13]). For each x- and y-value of each output point for a given image we estimate the standard deviation among the ensemble model outputs, and treat this as the epistemic uncertainty.

## 2.3 Combined uncertainty

For the aleatoric uncertainty, we computed the average $\log(b)$ tensors over the ensemble and used the exponential of this average as our $b$-values. We computed the component-wise total variance $V$ by adding the ensemble variance estimate to the Laplace variance $2b^2$. We then inverted the Laplace variance function to get a modified $\hat{b} = \sqrt{V/2}$, which was used to calculate confidence intervals according to the Laplace distribution.

## 3 Data sets

For training data only in-house annotation was used (details to be included after the anonymous review is finished). The images were sampled randomly from a set of mammograms from screening. All annotations were made by the first author, who has no formal background in radiology or radiography. The number of in-house annotated images was 545.

From the same source as the in-house training data we sampled a non-overlapping set of mammograms to be annotated by two radiographers. A total of 94 images were annotated by both.

In order to evaluate our models' generalizability, we also tested it on the external dataset InBreast (Moreira et al. [14]), which had 200 annotated images.

## 4 Experimental setup

The annotations of the pectoral muscle in the datasets were represented as a list of points along the border of the muscle. The number of points varied among the data sets and also among the images in each set. To facilitate the subsequent use of key points, the annotations were standardized to $n = 10$ equidistant points along the annotated path, where the first one was on the upper edge and the last one was on the vertical edge.

The HR-net was set up with a depth of 32, while the GCN modules that compute the $\Delta v$ and $b$ tensors had 6 layers and 256 filters. The number of coordinate iterations $T$ was set to 3. In line with Li et al. [7], We use the average of the ground truth

locations $v^*$ over the training set as starting values $v_0$.

The models were trained with the ADAM optimizer with a learning rate of 0.0001 and a batch size of 4 over 200 epochs, which was enough for overfitting the models. The cross validation used 5 folds, where each model was trained on 4 of them. The last one was used for monitoring the log-likelihood and parameters that gave the highest value on the validation fold was saved. This procedure gives inflated performance on the validation fold, but our purpose was only to create a model ensemble to be used on the test sets, not to cross-validate the models' performance on the training set.

Only a minimum of image preprocessing was performed. Right-side images were flipped, so that all images had the breast on the left side, with the pectoral muscle in the top left corner, if present. The images were resized to 512 x 512 pixels and the pixel values were re-scaled to [0, 1]. No data augmentation was used.

# 5 Results

## 5.1 Predictive performance

We measure the location errors by the average absolute difference between the predicted coordinates and the ground truth:

$$\frac{1}{2nM^{\text{test}}} \sum_{k}^{M^{\text{test}}} \sum_{i}^{n} \sum_{j \in \{x,y\}} |v_j^{*k,i} - \mu_j^{k,i}|$$

The location values $v^*$ and $\mu$ were scaled to the range $[0, 1]$, so the error estimates can be interpreted as fractions of the height and width of the images.

Recall that the internal test set had a ground truth annotation by two radiographers, and $v^*$ above was defined as the average of these. The model's average error on this data set was 0.0054, while the average error compared to each of the radiographers individually gave 0.0061 and 0.0062, respectively. For comparison, the average between-radiographer error was 0.0057, so the model performance is essentially on par with our human experts.

For the INBreast data set, the average error was higher, as would be expected due to the domain shift, but still quite acceptable: 0.0120.

## 5.2 Uncertainty estimates

The crucial property of uncertainty estimates is that they accurately model the empirical errors, so that the model can identify the cases where its location predictions should not be trusted. To test this we standardize the model errors by divided them with the predicted standard deviation. Under the model assumptions, these standardized errors should follow

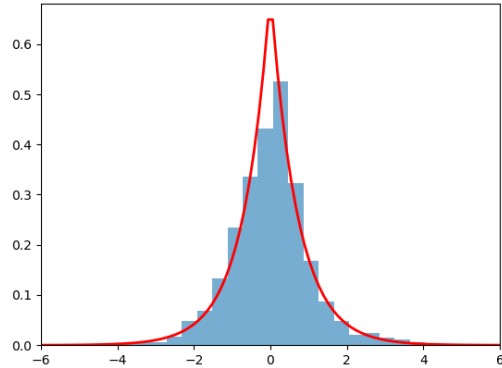

**Figure 3.** Histogram of standardized error distribution for the internal test set.

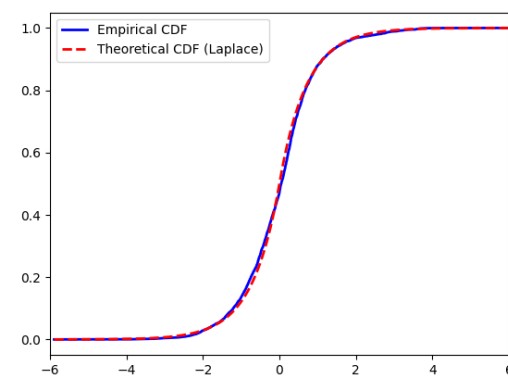

**Figure 4.** Comparison of the empirical and theoretical cumulative distribution for the internal test set.

the standard Laplace distribution. Figure 3 gives the histogram of these for the internal test set, together with the probability density function for the standard Laplace. Figure 4 shows a plot of the corresponding empirical cumulative distribution together with the theoretical one. These plots show a very strong correspondence. The maximum difference between the empirical and theoretical cumulative distributions is 0.0459, which confirms a very good match.

Figure 5 illustrates the model output on the two mammograms shown in the introduction, where we have zoomed in on the muscle. The red dots show the average annotation points of the two radiographers. The white and orange ellipses show the epistemic and aleatoric 99% confidence areas, while the yellow ellipses give the combined 99% confidence areas. As expected, the model predictions are more accurate for the left image with clearly visible muscle, and the small uncertainty ellipses confirm that the model is more certain about these. In the right image, we see that the model is more certain (and accurate) for

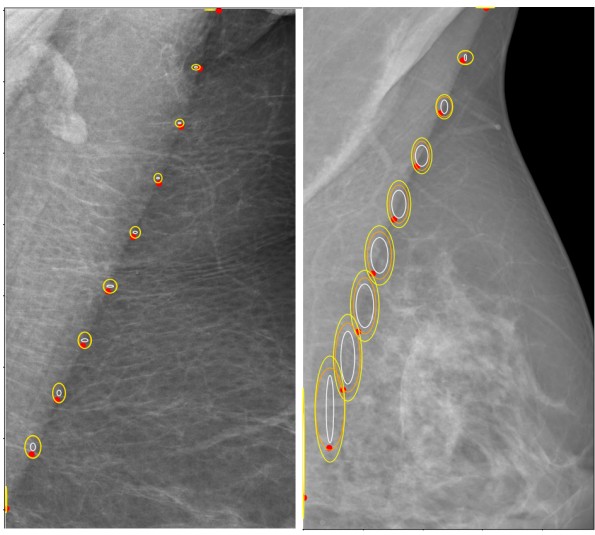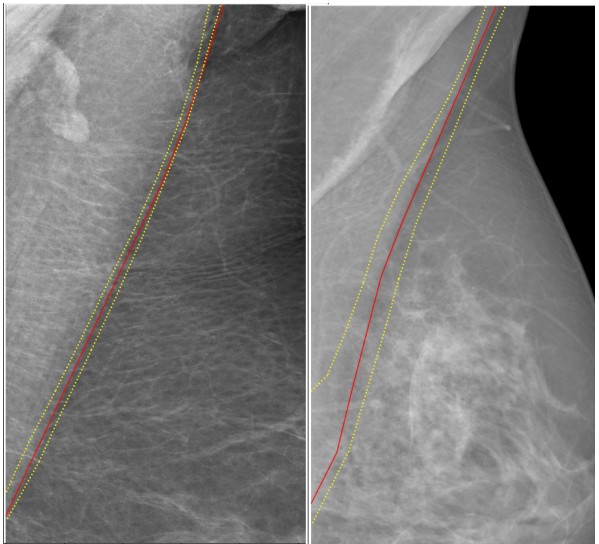

**Figure 5.** Sample mammograms with ground truth and uncertainty ellipses. The center of the ellipses are the predicted locations (not depicted)

**Figure 6.** Sample mammograms with ground truth and uncertainty bands.

the top points and more uncertain about the lower ones, which fits the visual impression. We also see that the epistemic uncertainties are smaller than the the aleatoric ones, which is desirable.

In Figure 6 the ellipses are converted to uncertainty bands (yellow dotted lines), while the red lines show the interpolated muscle annotations.

Figures 7 and 8 give the histogram and cumulative distribution plots for the external test set, which also shows an acceptable match with a maximum difference of 0.1248. We see that the empirical distribution is shifted to the left, compared to the standard Laplace. This is likely due to a slightly different annotation practice, in that the annotators of the external images may have included a larger part of the blurry areas. This is confirmed by visual inspection of the images, as illustrated in 9 where we have problems seeing the lower part of the annotated muscle. Here we have included the border predicted by our model in yellow, and we see a perfect match in the upper, more visible part of the muscle.

## 6 Discussion

We have successfully trained a GCN model to predict the border points of pectoral muscles, while simultaneously estimating the error distribution of these estimates, by interpreting the model outputs as the parameters of a Laplace distributions. The model's uncertainty estimates were remarkably accurate, in that the standardized errors followed the standard Laplace distribution almost perfectly. This means that the uncertainty estimates were almost perfectly calibrated out-of-the-box. A practical implication is that if we define a desired confidence level of, say 95%, the confidence intervals would in fact cover the

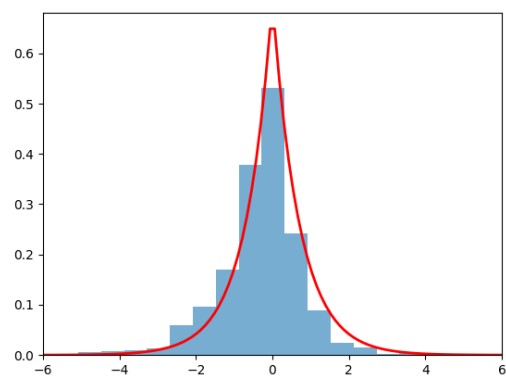

**Figure 7.** Histogram of standardized error distribution for the external test set.

ground truth in 95% of the cases. This will be very useful for later applications of the model, since it can reliably identify images that should be evaluated by a human expert. The model's point prediction performance on the test set was also convincing, with an L1-error similar to the difference between the two radiographers, which might be considered a lower bound on the possible performance. One might expect weaker performance because the training set was annotated by a non-professional, but this apparently made little difference. The model performed reasonably well on the external data set, despite substantial domain shift. We also suspect that the main reason for the weaker results may be a different annotation practice for uncertain cases.

Using the negative log-likelihood as a "loss function" may be unfamiliar to some, since it does not represent model errors directly, and is not even bounded downwards. This is not a problem, however,

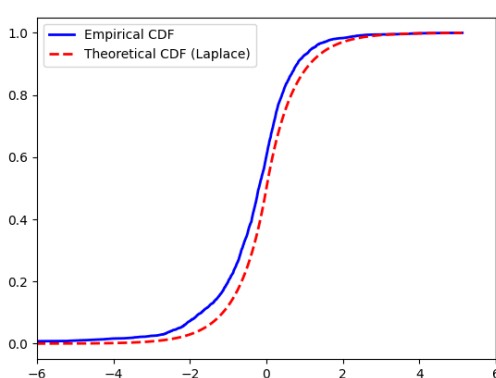

**Figure 8.** Comparison of the empirical and theoretical cumulative distribution for the external test set.

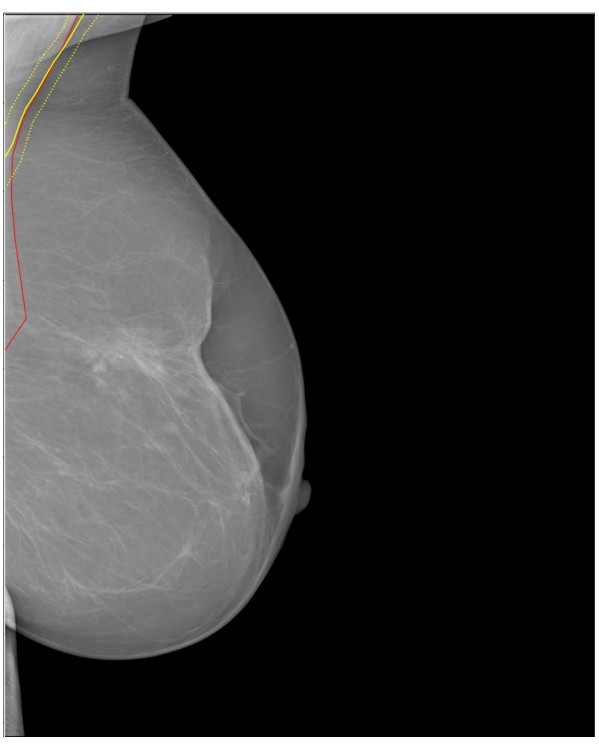

**Figure 9.** Example image from the INBreast test set, where the annotated boundary may seem large.

as long as it points the gradients in the direction that improves the model. The use of early stopping through monitoring of the log-likelihood on the validation folds was necessary, as over-fitting produces models that radically underestimate the aleatoric uncertainty, even though the predicted locations might still be good. Therefore, monitoring of the full log-likelihood was preferable to monitoring of the location errors.

# 7 Conclusion

In this work, we presented a novel GCN model for detecting pectoral muscle boundaries in mammograms, with integrated uncertainty estimation. By modeling both aleatoric and epistemic uncertainties, we were able to produce accurate predictions of key boundary points, alongside uncertainty estimates that can help identify cases requiring human review. Our approach achieved results on par with human annotators and demonstrated robustness across domain shifts, as shown in tests on the InBreast dataset.

The proposed uncertainty-aware framework has potential applications in clinical workflows, where the ability to flag uncertain cases could assist radiographers in prioritizing manual reviews. Future work will explore the integration of expert-annotated data to further refine the model and improve generalizability to diverse datasets, as well as investigate its practical use in a clinical setting.

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
