# OpenReview forum: "Graph convolutional neural networks with uncertainty modelling applied to edge detection in mammograms"
_NLDL.org/2025/Conference — Submitted to NLDL 2025_

### Official Review · Reviewer_jouS · 2024-09-24
**Well-written paper, clear contribtions**

**Confidence:** 3

**Summary:**

The paper proposes an uncertainty modeling approach based on graphc convolutional networks (GCN) on the application of idefntifying borders of the pectoral muscle in mammograms (in the medio-lateral oblique (MLO) view). The approach modifies a GCN to not only locate key points but also uncertainty estimates. The approach is benchmarked on an in-house dataset including different annotators and a public dataset.

**Strengths:**

- The paper is well-written and easy to follow along
- The application is well motivated and the approach addresses a relevant area of research
- The methods is benchmarked on a publicly available data set

**Weaknesses:**

- The experimental section confuses a little bit as the central motivation is about a reliable estimation of uncertainties - that's why the authors overfit the models. However, then also the section talks about the predictive and relating it to the human performance. It is unclear at certain points which models and/or configurations the authors talk about. Suggestion: make this clear from the beginning and introduce maybe some identifiers for the models that are being used.
- The in-house dataset was annotated by a non-expert. I cannot judge (in the light of the contirbution of the paper) how critical that actually is.

**Justification:**

First of all, I am not expert in this field. However, I think the paper presents some novelty worth sharing.

Questions:
- For the training the in-house data set was used - what are the results when you use the public data set?
- Lines 216-218: a bactch size of 4 seems to be quite low. Can you elaborate more on that design decision?
- Lines 241-250: this discussion seems a bit superficial. Is the overfitted model used for this discussion? And the test results are based on the training set? I think it is a bit sketchy to compare the predictive performance based on this experimental setup (moreover: "quite acceptable" - is there some king of guidance being available from medical doctors to justify such a statement?)

Some minor points:
- Line 55: Yu et al. appears twice
- Line 122: point after Li et al[.]
- Line 127: missing workd betwenn 'takes' and 'input'
- Line 129: \times instead of $x$

---

> ### Author Rebuttal · Authors · 2024-10-23
>
> Reviewer comment:
> The experimental section confuses a little bit as the central motivation is about a reliable estimation of uncertainties - that's why the authors overfit the models.
>
>
> Response:
> The models were not overfit. On the contrary, we use the validation fold to save the model that fits the best on that fold. There was a serious typo in the manuscript. In the sentence, the word “not” was incorrectly inserted, inverting the meaning: “The last one was not used for monitoring the log-likelihood and parameters that gave the highest value on the validation fold was saved.” We are sorry for this error, which will of course be corrected.
>
>
> Reviewer comment:
> However, then also the section talks about the predictive and relating it to the human performance. It is unclear at certain points which models and/or configurations the authors talk about. Suggestion: make this clear from the beginning and introduce maybe some identifiers for the models that are being used.
>
> Response:
> The comparison to human performance is all about comparisons to the radiographers’ annotations of the in-house test set.
>
> Reviewer comment:
> The in-house dataset was annotated by a non-expert. I cannot judge (in the light of the contirbution of the paper) how critical that actually is.
>
>
> Response:
> Our argument is that since the model learned well even from amateur annotations, it would likely do at least as well with professional annotation.
>
> Reviewer comment:
> For the training the in-house data set was used - what are the results when you use the public data set?
>
>
> Response:
> The evaluations of the model trained on inhouse-data on the external data set are given in Figures 6-7. We did not train the model on the external data.
>
> Reviewer comment:
> Lines 216-218: a bactch size of 4 seems to be quite low. Can you elaborate more on that design decision?
>
>
> Response:
> This was chosen to fit the GPU memory available.
>
> Reviewer comment:
> Lines 241-250: this discussion seems a bit superficial. Is the overfitted model used for this discussion? And the test results are based on the training set? I think it is a bit sketchy to compare the predictive performance based on this experimental setup (moreover: "quite acceptable" - is there some king of guidance being available from medical doctors to justify such a statement?)
>
>
> Response:
> This would maybe be clarified by the fact that the model was not overfit? Our argument for the quality of the point predictions is that they differ from the expert predictions about as much as the experts differ between them. However, the main point is that the model’s estimated error distribution fits the theoretical on very well, which implies that it can be used to create confidence intervals that are close to perfectly calibrated.
>
> Reviewer comment:
> Line 55: Yu et al. appears twice
> Line 122: point after Li et al[.]
> Line 127: missing workd betwenn 'takes' and 'input'
> Line 129: \times instead of x
>
>
> Response:
> Thanks!

---

### Official Review · Reviewer_zV6X · 2024-10-08
**Interesting paper, needs more elaboration**

**Confidence:** 3

**Summary:**

The paper presents a novel approach to uncertainty estimation for the detection of pectoral muscle boundaries in mammograms. Pectoral muscle can fail to contrast strongly with breast tissue in mammograms, leading to uncertainty among medical experts about the ground-truth delineation. This raises the question of how to represent uncertainty for the task.

The novelty comes from the application of graph convolutional neural networks not only to landmark detection but also to the representation of aleatoric and epistemic uncertainty for the landmark locations. The paper views landmark data as inherently random (i.e. aleatoric) and thus seeks to model it as such. Two GCNs serve this purpose: one which locates landmark points and another which models aleatoric uncertainty using a product of Laplace distributions, one for each coordinate of each landmark point in each image. They view this product as a likelihood function and optimize the aleatoric GCN model model via maximum likelihood estimation on the parameters. Further, the paper uses an ensemble of models, presumably also GCNs, to model the epistemic uncertainty.

Experiments were performed on in-house mammogram data and validated on publicly available mammogram data. An ensemble of GCNs was produced via five distinct training runs on a five-fold split of the data. The ensemble produces an average error close to the average error between radiographers on a test set held out from the in-house mammogram data. The average error on the publicly available mammogram data was double that of the error on the in-house test set. Model uncertainties were validated by comparing the empirical distribution of test scores against a one-dimensional Laplace distribution and by visual inspection.

**Strengths:**

The task is clearly explained, and the motivation of the paper is almost immediately apparent, providing this paper with clear context for a reading. The figures give a good illustration of the problem, which is to localise the muscle even when an acquired image does not clearly resolve the muscle. The use of uncertainty ellipses provides, in my opinion, an intuitive representation of uncertainty about the location of landmarks which delineate the pectoral music boundary, and I speculate that this would be useful in the clinical setting.

The methods section is the nicest part of the paper, with well-chosen references to aid further understanding.

**Weaknesses:**

The introduction does not adequately motivate the use of graph convolutional neural networks for the problem. Skimming the methods section shows that the motivation appears there. This should be moved to the introduction, or at least hinted at, with further elaboration in the methods section.

The paper would certainly benefit from a figure which depicts the model pipeline. There are multiple GCNs which are applied to the task, which is not obvious when skimming the paper.

The data section is too short and is unclear. The paper is very short on details about the training data. In particular, it's not clear what the training data are, aside that they are mammograms. I assume that the 94 images annotated by radiographers comprise the test set. If so, this should be stated explicitly. Moreover, the use of an exclusively private dataset for training poses problems for reproducibility.

The experiment section is also unclear and even contradictory in places. For example, when describing the development of the model ensemble using the five-fold split, one reads “The last one was not used for monitoring the log-likelihood and parameters that gave the highest value on the validation fold was saved.“ Obviously, you can’t save model parameters based on the performance of the validation set unless the loss function is monitored on the validation set. I also would not use the term "cross-validation", since the held-out folds are used for parameter selection to produce the model ensemble.

As stated in the results section, landmark coordinates were rescaled to the unit interval, giving their L1 differences an interpretation as percentages of the height and width of the images. This is meaningful only if the relative area occupied by the pectoral muscle has low variance across all images. However, the paper does not address this. Are there images where the foreground is large and other images where the foreground is small? Were the data processed to control for this? Since the paper states that the image pre-processing itself was minimal, this is a concern.

I would not assert that the model’s performance is on par with human experts just because the errors are roughly the same as the average between-radiographer error on the internal test set. The numbers on the external test set show clearly that this is not the case, and the assertion is therefore misleading.

“… we standardize the model errors by divided them with the predicted standard deviation.” I think the authors meant that they subtract the mean and then divide by the standard deviation. Just dividing by the standard deviation gives numerical values greater than or equal to zero, which is not consistent with the depictions in Figure 2 and Figure 3.

The authors compute the Kolmogorov–Smirnov statistic when comparing the empirical cumulative distribution function to the theoretical standard Laplace distribution function. However, they don’t use this statistic in a hypothesis test, namely for the Kolmogorov-Smirnov test. Why not?

**Justification:**

This paper seems to make a novel contribution, but it requires revision to communicate this contribution more clearly.

---

> ### Author Rebuttal · Authors · 2024-10-23
>
> Reviewer comment:
> The introduction does not adequately motivate the use of graph convolutional neural networks for the problem. Skimming the methods section shows that the motivation appears there. This should be moved to the introduction, or at least hinted at, with further elaboration in the methods section.
>
> Response:
> Agreed, we have restructured this a bit.
>
> Reviewer comment:
> The paper would certainly benefit from a figure which depicts the model pipeline. There are multiple GCNs which are applied to the task, which is not obvious when skimming the paper.
>
> Response:
> A very good point, and we have added such a figure.
>
> Reviewer comment:
> The data section is too short and is unclear. The paper is very short on details about the training data. In particular, it's not clear what the training data are, aside that they are mammograms. I assume that the 94 images annotated by radiographers comprise the test set. If so, this should be stated explicitly. Moreover, the use of an exclusively private dataset for training poses problems for reproducibility.
>
> Response:
> We plan to include more information on the data, including its source, but have kept this out for now to keep the manuscript anonymous during the review process.
>
> Reviewer comment:
> The experiment section is also unclear and even contradictory in places. For example, when describing the development of the model ensemble using the five-fold split, one reads “The last one was not used for monitoring the log-likelihood and parameters that gave the highest value on the validation fold was saved.“ Obviously, you can’t save model parameters based on the performance of the validation set unless the loss function is monitored on the validation set. I also would not use the term "cross-validation", since the held-out folds are used for parameter selection to produce the model ensemble.
>
> Response:
> Thanks, nice catch of this typo! It should read “The last one was used …”. We also agree that "cross-validation" is somewhat inaccurate, but decided that this phrasing was the fastest way to communicate the procedure. We think it is ok as long as we don’t quote CV performance.
>
> Reviewer comment:
> As stated in the results section, landmark coordinates were rescaled to the unit interval, giving their L1 differences an interpretation as percentages of the height and width of the images. This is meaningful only if the relative area occupied by the pectoral muscle has low variance across all images. However, the paper does not address this. Are there images where the foreground is large and other images where the foreground is small? Were the data processed to control for this? Since the paper states that the image pre-processing itself was minimal, this is a concern.
>
> Response:
> We assume the reviewer argues that the relative error in the implied size of the muscle is the relevant measure. We disagree on this and will argue that the absolute error is more relevant. Some of the images have a very small visible muscle, and in some it is missing completely (in which case the ground truth places all points in the top left corner). For the intended use as image quality control these are similar, which makes the absolute error more relevant. Incidentally, we see this as a strength of our methodology compared to segmentation studies that often use the intersection-over-union metric, which is sensitive to small errors in small objects and is even undefined for missing objects.
>
> Reviewer comment:
> I would not assert that the model’s performance is on par with human experts just because the errors are roughly the same as the average between-radiographer error on the internal test set. The numbers on the external test set show clearly that this is not the case, and the assertion is therefore misleading.
>
> Response:
> We disagree on this. The model performs on par with the human experts on an independent test set from the same image distribution. The fact that it performs worse on an external data set, which is based on scanned images rather than digital ones (a fact admittedly not mention in our manuscript) should be attributed to the domain shift. We also observe that some of the images in the external data set show an annotation practice of annotating larger parts of blurry areas, which corresponds to the horizontal shift in the histogram/cdf figures 6-7 (currently 7-8).
>
> Reviewer comment:
> “… we standardize the model errors by divided them with the predicted standard deviation.” I think the authors meant that they subtract the mean and then divide by the standard deviation. Just dividing by the standard deviation gives numerical values greater than or equal to zero, which is not consistent with the depictions in Figure 2 and Figure 3.
>
> Response:
> No, we meant what we wrote there. The x-axis in these figures go from -6 to 6 and do represent positive and negative differences between the ground truth and the predictions. In the corresponding figures 4-5 for the InBreast data, you see a clear left-shift of the histogram and cdf, which correspond to the fact that the ground truth accounts for larger parts of the blurry images as mentioned above (making both the x- and y-values larger).
>
> Reviewer comment:
> The authors compute the Kolmogorov–Smirnov statistic when comparing the empirical cumulative distribution function to the theoretical standard Laplace distribution function. However, they don’t use this statistic in a hypothesis test, namely for the Kolmogorov-Smirnov test. Why not?
>
> Response:
> We have 20 data points for 94 images, so n = 1880, which is so high that the p-value is low even though the distributions are similar. The K-S test is useful for testing whether a data set is generated exactly from a specific distribution, but not for measuring the distance between distributions, which is our concern in this application.

---

### Official Review · Reviewer_ZQmc · 2024-10-09
**Review of paper on GCNs with uncertainty modelling applied to pectoral muscle detection in mammograms**

**Confidence:** 4

**Summary:**

This paper presents a method for identification and uncertainty estimation of pectoral muscle border in X-ray mammogram images using graph convolutional neural networks. The problem of identifying the border of pectoral muscle on X-ray mammogram images is a crucial step in identification of potential breast cancer indicators, but due to the variability in muscle visibility across images, significant uncertainty exists, which this paper aims to address. The proposed method was trained on in-house data, and was tested on both the in-house data annotated by two radiographers as well on the external data, and has shown promising initial results. The uncertainty estimations provide a potential method for identification of X-ray mammogram images which require additional human review/intervention.

**Strengths:**

Paper has the following strengths:
- The paper is written in clear and concise manner and was easy to follow. Aleatoric and epistemic uncertainty estimation (as main focuses of the paper) were explained in detail, while the background of the problem was simple to understand and did not require medical expertise. Additionally, visualization of images, along with pectoral muscle border predictions and uncertainty estimations, made the paper even more straightforward to understand.
- A novel method for identification and uncertainty estimation of pectoral muscle border in X-ray mammogram images using graph convolutional neural networks was introduced, and has shown promising initial results.
- Method's generalizability was tested on external data.

**Weaknesses:**

The following weaknesses of the paper should be addressed:
- $\textbf{Introduction}$. Introduction should contain more detail on clinical applications of pectoral muscle identification: Why is it important (e.g., because it can overlap with fibroglandular tissue and thus needs to be excluded from quantitative analysis of breast parenchyma)? How is it being performed in hospitals nowadays, and what are the drawbacks? What is the radiologist/radiographers performance? How does your method address current issues? Additionally, more references for first paragraph of introduction are required.
- $\textbf{Introduction Cont'd}$. There have been numerous studies (especially over the last five years) in deep learning application to mammography screening, and particularly in pectoral muscle segmentation (e.g., On Segmentation of Pectoral Muscle in Digital Mammograms by Means of Deep Learning by H. Soleimani et al., Deep learning based pectoral muscle segmentation on Mammographic Image Analysis Society (MIAS) mammograms by Y. J. Kim, et al.), while you reference only two. How does your work differ from the previous methods?
- $\textbf{Method}$. More detailed description of GCN method from Li et al. is required (e.g., what is the architecture of HR-Net, overview of DAG, etc.). Why do you skip the global step? How can you assume that all components (images and key points) are independent when computing likelihood of Laplace distribution? Have you tried using Normal distribution (a comparison of results between Laplace and Normal distribution of key points would have been a nice addition)?
- $\textbf{Data sets}$ Significantly more details are required when describing the internal data set: Where was it acquired, and how? Demographic information, scanner information, annotation protocol? Why wasn't the training data annotated by experienced radiologists/radiographers? Why didn't you make use of other available X-ray mammography open-source datasets (e.g., the NYU breast cancer screening dataset, CSAW-CC, VinDr-Mammo, etc.) to either enrich your training data (as you are working in significantly low data regime), or to perform additional testing of your method?
- $\textbf{Experimental setup}$ Additional justification on some hyper-parameters would be beneficial: Why did you select T=3 for the total number of iterations (did you try higher values of T, and perhaps notice no significant performance gains)? Why did you select n=10 for the number of points?
- $\textbf{Results}$ You could further describe the meaning of error values on test data (e.g., error of 0.012 corresponds to average predicted point coordinate being 512*0.012=6.144 pixels away from the ground truth point coordinate. How can being over 6 pixels in both x and y dimension away from pectoral muscle on average be considered a good performance?). A large drawback of this paper is missing subsection on comparison of your method to state-of-the-art methods, especially for InBreast data which has been used extensively. Is your method comparable (or non-inferior) in identification of pectoral muscle to state-of-the-art ones, with an addition of uncertainty estimation? Or is the identification of pectoral muscle significantly worse?
- $\textbf{Conclusion}$ How can this method be used in clinical settings? How would radiologists/radiographers benefit? Standard practice nowadays is for at least one medical expert (and more often, two) to review mammogram images. How does your method affect this? How will you evaluate your method even further?

**Final Rebuttal Confidence:**

4

**Final Rebuttal Justification:**

After the rebuttal period, the updated version of the manuscript, as well as other reviewers' and author's comments, the rating and the main sentiment of the previous revision remains.

The paper has introduced a novel method for identification and uncertainty estimation of pectoral muscle in X-ray mammogram images using graph convolutional neural networks. While the main idea of the paper was clearly described and the method has shown somewhat promising initial results on the in-house test data, there are a number of drawbacks which need to be addressed. While it was a well-thought-out idea and approach, it still requires additional work:

- more details and discussions on the clinical impact of the method,
- more details and discussions on the state-of-the-art methods, as well as on other approaches,
- more details and discussions on the in-house data, as well as additional testing.

**Justification:**

The paper has introduced a novel method for identification and uncertainty estimation of pectoral muscle in X-ray mammogram images using graph convolutional neural networks. While the main idea of the paper was clearly described and the method has shown promising initial results, there are a number of drawbacks which need to be addressed prior to paper publication. While it was a well-thought-out idea and approach, it still requires additional work:
- more detail on the clinical application and impact of the method,
- more detail on the background on the problem and discussion of the state-of-the-art methods,
- more detailed description of the method, and not just referring to the paper which introduced it,
- more details on the in-house data, and additional testing and discussion of the results.

If the above-mentioned drawbacks were to be addressed in the future, the paper would be accepted. However, for the time being, it is rejected.

---

> ### Author Rebuttal · Authors · 2024-10-23
>
> Reviewer comment:
> Introduction. Introduction should contain more detail on clinical applications of pectoral muscle identification: Why is it important (e.g., because it can overlap with fibroglandular tissue and thus needs to be excluded from quantitative analysis of breast parenchyma)? How is it being performed in hospitals nowadays, and what are the drawbacks? What is the radiologist/radiographers performance? How does your method address current issues?
>
>
> Response:
> We try to strengthen this aspect in the new version.
>
>
> Reviewer comment:
> Additionally, more references for first paragraph of introduction are required.
>
>
> Response:
> We have tried to strengthen the literature review part.
>
> Reviewer comment:
> Introduction Cont’d. There have been numerous studies (especially over the last five years) in deep learning application to mammography screening, and particularly in pectoral muscle segmentation (e.g., On Segmentation of Pectoral Muscle in Digital Mammograms by Means of Deep Learning by H. Soleimani et al., Deep learning based pectoral muscle segmentation on Mammographic Image Analysis Society (MIAS) mammograms by Y. J. Kim, et al.), while you reference only two. How does your work differ from the previous methods?
>
>
> Response:
> We have tried to improve this.
>
> Reviewer comment:
> Method. More detailed description of GCN method from Li et al. is required (e.g., what is the architecture of HR-Net, overview of DAG, etc.).
>
>
> Response:
> We have included a section on the computation pipeline of our model, which hopefully covers some of this.
>
>
> Reviewer comment:
> Why do you skip the global step?
>
>
> Response:
> We tried this as an Occam’s razor / ablation step and found that our model even benefited slightly from switching the global step to an additional local one, so we decided to go for the simpler solution. (As an aside, there is a problem with Li et al’s description of the perspective transform, which actually uses only 6 of the 8 degrees of freedom.) The reason why we can do without the global step may be that the starting points (the average coordinates in the training set) are relatively close to the final points in most of our images. We will mention this in the discussion.
>
>
> Reviewer comment:
> How can you assume that all components (images and key points) are independent when computing likelihood of Laplace distribution?
>
>
> Response:
> Again this is an Occam’s razor approach where we try a simple model and stick with it when it fits data well.  A reasonable next step would be to include correlations between the x and y of each point as model output. However, including a full covariance matrix for all 20 dimensions would probably not be a good idea.
>
>
> Reviewer comment:
> Have you tried using Normal distribution (a comparison of results between Laplace and Normal distribution of key points would have been a nice addition)?
>
>
> Response:
> Yes, we tried this, and got slightly worse results. We would like to include this, but again have the page limit. Although the normal distribution would be a more ‘normal’ choice in most cases, the Laplace is a natural default here, because it generalizes the L1-norm used by Li et al.
>
> Reviewer comment:
> Data sets Significantly more details are required when describing the internal data set: Where was it acquired, and how? Demographic information, scanner information, annotation protocol?
>
>
> Response:
> We plan to include more on this, including the source of the data, but exclude it for now, to keep the anonymity of the manuscript in the review phase.
>
>
> Reviewer comment:
> Why wasn't the training data annotated by experienced radiologists/radiographers? Why didn't you make use of other available X-ray mammography open-source datasets (e.g., the NYU breast cancer screening dataset, CSAW-CC, VinDr-Mammo, etc.) to either enrich your training data (as you are working in significantly low data regime), or to perform additional testing of your method?
>
>
> Response:
> The internal training annotations were used due to the easier access. Radiographer time was a more limited resource, and therefore was used only for the test set.  One can always do more, and this was what we had time and resources for in our current project.
>
> Reviewer comment:
> Experimental setup Additional justification on some hyper-parameters would be beneficial: Why did you select T=3 for the total number of iterations (did you try higher values of T, and perhaps notice no significant performance gains)? Why did you select n=10 for the number of points?
>
>
> Response:
> We did not perform any systematic grid search for these parameters, but some initial unsystematic experiments indicated that these we reasonable choices.
>
> Reviewer comment:
> Results You could further describe the meaning of error values on test data (e.g., error of 0.012 corresponds to average predicted point coordinate being 512*0.012=6.144 pixels away from the ground truth point coordinate. How can being over 6 pixels in both x and y dimension away from pectoral muscle on average be considered a good performance?).
>
>
> Response:
> We do not agree that the number of pixels is a relevant measure. If one e.g. doubles the resolution, the image appears visually similar, while the pixel distances double. The error is smaller for images with a sharp muscle boundary and larger for blurry images. Anyway, when we get errors that are similar to between-radiographer discrepancies, we argue that this is a good performance.
> Anyway, the main thrust of the paper is the uncertainty modelling, and the fact that the standardized empirical model errors on the test set match the standard Laplace distribution almost perfectly is the main results. This does e.g. imply that when we use the uncertainty estimates to define confidence intervals, these will have close to the desired coverage (e.g. a 95% CI will in fact cover 95% of the ground truth values). We include some discussion of this.
>
>
> Reviewer comment:
> A large drawback of this paper is missing subsection on comparison of your method to state-of-the-art methods, especially for InBreast data which has been used extensively. Is your method comparable (or non-inferior) in identification of pectoral muscle to state-of-the-art ones, with an addition of uncertainty estimation? Or is the identification of pectoral muscle significantly worse?
>
>
> Response:
> The published works we have found use segmentation methods and evaluate in terms of intersection-over-union (IOU), which is different from our sequence-of-points approach.
>
> Reviewer comment:
> Conclusion How can this method be used in clinical settings? How would radiologists/radiographers benefit? Standard practice nowadays is for at least one medical expert (and more often, two) to review mammogram images. How does your method affect this? How will you evaluate your method even further?
>
>
> Response:
> The pectoral muscle detection step is a part of the image quality evaluation which comes before the radiologist evaluation. We will include some information on this in a later revision.
>
>
> Reviewer comment:
> The paper has introduced a novel method for identification and uncertainty estimation of pectoral muscle in X-ray mammogram images using graph convolutional neural networks. While the main idea of the paper was clearly described and the method has shown promising initial results, there are a number of drawbacks which need to be addressed prior to paper publication. While it was a well-thought-out idea and approach, it still requires additional work:
> •	more detail on the clinical application and impact of the method,
> •	more detail on the background on the problem and discussion of the state-of-the-art methods,
> •	more detailed description of the method, and not just referring to the paper which introduced it,
> •	more details on the in-house data, and additional testing and discussion of the results.
> If the above-mentioned drawbacks were to be addressed in the future, the paper would be accepted. However, for the time being, it is rejected.
>
>
> Response:
> We have tried to strengthen these points in the new version, and will improve it further for a later version.

---

### Official Review · Reviewer_6uuY · 2024-10-11
**Graph convolutional neural networks with uncertainty modelling applied to edge detection in mammograms**

**Confidence:** 4

**Summary:**

The paper addresses the problem of delineating the border of pectoral muscle in mammograms. The authors present a modification of the graph convolutional network that locates points along this boundary and also provides uncertainty estimates. They test the method on one publically available dataset and one in-house dateset. From the presented results, it is difficult to assess the value of the proposed approach.

**Strengths:**

The problem is relevant, and the approach using graph convolutional networks is well-motivated. The writing is clear, and the work is well presented.

**Weaknesses:**

A large part of the paper deals with uncertainty estimation.  Almost all text in the Method section is about uncertainty. However, the results of the uncertainty estimation are not convincing. Maybe this is just the way the results are presented, but it is difficult to see the value of the suggested method.

To write just some concerns regarding this, here are the questions about Figure 4: The red dots are averages of two annotations. It would help to understand the benefits of the method, if we could see those two annotations. Why are the dots not in the middle of the confidence areas? Why are all ellipses axis-aligned and either elongated in the x or y direction? Are we not mostly interested in uncertainty in the normal direction, as the displacement in the tangential direction only changes parametrization, but not the resulting curve? In Figure 5, why is the ground truth outside the uncertainty band for the left image?

In general, the benefits of the method are not evident. We hear nothing about what the alternatives are. So, despite mentioning some earlier works, the proposed method is not placed in the state of the art. There is no quantification of the method, and no comparison.

The discussion and conclusion simply ignore those problems and provide no explanation. They claim to produce accurate results — but it’s unclear what accurate is in this context. They also claim that uncertainty estimates can help identify difficult cases, but it is unclear to see, how this should work.

**Final Rebuttal Confidence:**

4

**Final Rebuttal Justification:**

I appreciate the rebuttal, and I also recognize that whether something is convincing, or not convincing, is somewhat subjective.

My most important concern was the lack of clarity: How good are the results? What do the results enable? What can I accomplish using this method, which I could not accomplish without it?

The rebuttal says that this is obvious because histograms and curves fit well. But what does 'fit well' mean in this context? What would "fit badly" be? I understand that you have achieved a fit of a certain quality. But I still don't know how is this quality better than anything else. I do understand that the method is somewhat unique. Still, the authors should be able to produce a naive baseline. Or naive translation of segmentation results into curve-based results.

**Justification:**

The paper proposes an approach that yields some results. However, there is no evidence for the quality of these results.

---

> ### Author Rebuttal · Authors · 2024-10-23
>
> Reviewer comment:
> A large part of the paper deals with uncertainty estimation. Almost all text in the Method section is about uncertainty. However, the results of the uncertainty estimation are not convincing. Maybe this is just the way the results are presented, but it is difficult to see the value of the suggested method.
>
> Response:
> This is covered by the histograms and cumulative distribution curves together with the estimated maximum difference in the latter. It may not have been clear enough in the manuscript, but if the empirical distribution displayed in these figures had a perfect match with the theoretical curve, it would mean that the uncertainty estimates were perfectly calibrated, so that if you create 95% confidence intervals, they would in fact cover exactly 95% of the ground truth values. This is the ideal situation. Given that the histograms and curves fits very well, the uncertainty modelling has been successful.
>
> Reviewer comment:
> To write just some concerns regarding this, here are the questions about Figure 4: The red dots are averages of two annotations. It would help to understand the benefits of the method, if we could see those two annotations. Why are the dots not in the middle of the confidence areas?
>
> Response:
> The red dots represent the ground truths for the test set, which are of course unknown to the model. The centres of the ellipses are the model’s point predictions. We considered displaying these as well, but found that the images became too crowded. We believe the reader can identify the centres without us marking them explicitly.
>
> Reviewer comment:
> Why are all ellipses axis-aligned and either elongated in the x or y direction? Are we not mostly interested in uncertainty in the normal direction, as the displacement in the tangential direction only changes parametrization, but not the resulting curve?
>
> Response:
> A very good point. The ellipses are all x-y oriented due to the assumption of independence of the error terms. We might include a correlation term in the model’s output to represent this, but unfortunately we will not be able to handle this in the present article.
>
> Reviewer comment:
> In Figure 5, why is the ground truth outside the uncertainty band for the left image?
>
> Response:
> It is supposed to be outside the uncertainty band with the frequency set in the confidence interval percentage. If they were always inside the ellipses, the method would not have done its job properly.
>
> Reviewer comment:
> In general, the benefits of the method are not evident. We hear nothing about what the alternatives are. So, despite mentioning some earlier works, the proposed method is not placed in the state of the art. There is no quantification of the method, and no comparison.
>
> Response:
> We are not aware of any published GCN model for coordinate estimation that gives uncertainty estimates, so direct comparisons are not easy to make, but we will include an extended review of the literature.
>
> Reviewer comment:
> The discussion and conclusion simply ignore those problems and provide no explanation. They claim to produce accurate results — but it’s unclear what accurate is in this context. They also claim that uncertainty estimates can help identify difficult cases, but it is unclear to see, how this should work.
>
> Response:
> This maps directly to the histograms and cdf’s that compare to the ideal Laplace distribution. The fact that the distribution matches so well implies that the model can be trusted to estimate its own uncertainty. This does e.g. imply that when we use the uncertainty estimates to define confidence intervals, these will have close to the desired coverage (e.g. a 95% CI will in fact cover 95% of the ground truth values). We will emphasize this more in the final version.
>
> Reviewer comment:
> The paper proposes an approach that yields some results. However, there is no evidence for the quality of these results.
>
> Response:
> Again, this view by the reviewer is probably due to a misunderstanding regarding the presented comparisons between the distribution, as discussed above.

---

### Meta-Review · Area_Chair_mNFS · 2024-11-02

**Recommendation:** Reject
**Confidence:** 4

**Metareview:**

This paper addresses the problem of uncertainty quantification for delineation of the pectoral muscle in mammograms. This is a highly interesting problem, and the reviewers find the paper well written and the problem well motivated, and they appreciate the external datasets used to evaluate the method.

However, the concerns voiced by several reviewers regarding the experimental validation and to some degree also the motivation for the choice of method make it hard to accept the paper in its current form. Therefore, unfortunately, I cannot recommend acceptance.

However, I highly encourage the authors to further develop the experimental validation, especially of the UQ part. To this end, you might find the recent paper [1] below helpful, which suggests different strategies for validating different types of segmentation uncertainty, which is highly related to the proposed problem.

[1] Kahl, Kim-Celine, et al. "ValUES: A Framework for Systematic Validation of Uncertainty Estimation in Semantic Segmentation." The Twelfth International Conference on Learning Representations.

**Suggested Changes To The Recommendation:**

3: I agree that the recommendation could be moved up

---

### Decision · Program_Chairs · 2024-11-06

Reject